# Cardiovascular risk factors and kidney function among automobile mechanic and their association with serum heavy metals in Southwest Nigeria: A cross-sectional study

Oluseyi Ademola Adejumo[1]*, Adenike Christianah Enikuomehin[1], Adeyemi Ogunleye[2], Walter Bamikole Osungbemiro[3], Alex Adedotun Adelosoye[4], Ayodeji Akinwumi Akinbodewa[1], Olutoyin Morenike Lawal[1], Stanley Chidozie Ngoka[5], Oladimeji Adedeji Junaid[1], Kenechukwu Okonkwo[1], Emmanuel Oladimeji Alli[1], Rasheed Olanshile Oloyede[1]

1 Department of Internal Medicine, University of Medical Sciences, Ondo City, Ondo State, Nigeria,
2 Department of Medical Laboratory Science, University of Medical Sciences, Ondo City, Ondo State, Nigeria, 3 Department of Chemistry, University of Medical Sciences, Ondo City, Ondo State, Nigeria, 4 Department of Family Medicine, University of Medical Sciences Teaching Hospital, Akure, Ondo State, Nigeria, 5 Department of Internal Medicine, Federal University Teaching Hospital, Owerri, Imo State, Nigeria

* oluseyiadejumo2017@gmail.com

## Abstract

### Introduction

The burden of cardiovascular disease (CVD) is huge due to its associated morbidity, mortality and adverse socio-economic impact. Environmental pollution as a risk factor contributes significantly to the burden of CVD, especially in the low and middle income countries. One of the effective strategies to reduce CVD burden is to prevent or detect cardiovascular risk factors early in at-risk population. This study determined some cardiovascular risk factors, kidney function, and their association with heavy metals among automobile mechanics.

### Method

This was a cross-sectional study involving 162 automobile mechanics and 81 age and sex matched controls. Serum levels of lead, cadmium and some cardiovascular risks were assessed and compared in the two groups. Associations between serum lead, cadmium and some cardiovascular risks were determined using correlation analysis. P value of <0.05 was taken as significant.

### Results

The mean ages of the automobile mechanics and controls were 47.27±9.99 years and 48.94±10.34 years, respectively. The prevalence of elevated serum cadmium was significantly higher in the automobile mechanics (25.9% vs 7.9%; p = <0.001). The significant cardiovascular risk factors in the automobile mechanics vs controls were elevated total cholesterol (32.1% vs 18.5%; p = 0.017), hyperuricemia (20.4% vs 1.2%; p = <0.001),

**Data Availability Statement:** All relevant data are within the paper and its Supporting Information files.

**Funding:** OAA UNIMED TETFUND IBFR 001 Institution Based Research Fund (IBRF) Tertiary Education Trust Fund The funders had no role in study design, data collection and analysis, decision to publish, or preparation of the manuscript.

**Competing interests:** The authors have declared that no competing interests exist.

elevated blood glucose (16.0% vs 4.9% p = 0.013); and alcohol use (55.1% vs 30.0%; p = 0.001). Among the automobile mechanics, there were significant positive correlations between serum cadmium, atherogenic index of plasma (AIP) (p = 0.024; r = 0.382) and tri-glyceride (p = 0.020; r = 0.391). Significant positive correlation was found between serum lead and neutrophil gelatinase associated lipocalin (NGAL) (p = <0.001; r = 0.329). There were significant positive correlation between serum cadmium level, AIP (p = 0.016; r = 0.373) and TG (p = 0.004; r = 0.439); between serum lead and NGAL in all the study participants (p = 0.005; r = 0.206).

## Conclusion

Automobile mechanics have notable exposure to heavy metals and a higher prevalence of some cardiovascular risk factors. Health education and sensitisation as well as policies that would regulate exposure of persons to heavy metals should be implemented in Nigeria.

## Introduction

Cardiovascular disease (CVD) is a disease of public health importance due to its increasing prevalence, associated morbidity, mortality and socio-economic impact [1–3]. CVD accounts for 17.9 million deaths globally. The burden of CVD is however, higher in low and middle-income countries (LMIC) [1, 2, 4]. Cardiovascular risk factors which account for CVD can be either traditional or novel. These risk factors include hypertension, diabetes mellitus, tobacco use, dyslipidaemia, obesity, depression, environmental pollution, low education, low grip strength and poor diet [1, 4]. The contribution of cardiovascular risk factors as drivers of cardiovascular morbidities and mortality varies in different countries [1, 4]. Prevention or early detection of cardiovascular risk factors constitute one of the major strategies to reduce the burden of CVD. Of the risk factors stated above, environmental pollution and contamination is a more common contributor to CVD in LMIC compared to high income countries [1, 4]. Occupational practices of artisans such as automobile mechanics may contribute to environmental pollution and contamination [5–7]. Previous studies within and outside Nigeria have reported higher serum levels of some heavy metals in automobile mechanics [8–10]. Hazardous occupational practices such as regular use of diesel and petrol to wash the hands and feet as well as constant oral sucking of fuel may partly account for higher levels of some heavy metals in automobile mechanics [11]. The report of Akpoveta et al [12] specifically showed higher concentration of heavy metals such as lead in diesel and petrol in Nigeria beyond the generally permissible levels in refined petroleum products.

Although, the pathophysiology of association between heavy metal exposure and CVD is not well understood, both experimental and human studies have implicated the activation of renin-angiotensin-aldosterone system, generation of reactive oxygen species and inflammation [13]. Heavy metals have also been found to be associated with increased prevalence of some cardiovascular risk factors such hypertension, hyperuricaemia and dyslipidaemia [13].

There is limited information on the relationship between heavy metals, cardiovascular risk and kidney function among those that are occupationally exposed in Nigeria. Most screening programs that aimed at reducing cardiovascular disease and kidney disease burden tend to pay less attention of those whose occupation increases their risk of developing these diseases. This study aimed to fill part of this existing gap by determining some cardiovascular risk factors,

kidney function, and their association with heavy metals among automobile mechanics. The findings of this study will also provide scientific evidence to advocate for safe occupational practices among automobile mechanics.

## Methodology

This was a cross-sectional study that was carried out in Ondo State, Nigeria between November 2021 and February, 2022. Ondo state is one of the 36 states in Nigeria and is located in the Southwestern part of the country.

### Study population

The study participants comprised of automobile mechanics in Ondo and Akure city, Ondo State. Inclusion criteria were automobile mechanic with a minimum working experience of 1 year, aged $\geq$ 18 years, and those who gave informed consent. Any automobile mechanic with known kidney disease, heart disease or liver disease based on history and physical examination was excluded from participating in the study. Controls were individuals who were not automobile mechanics and have not had significant exposure to petrochemicals.

### Sample size determination

The sample size was determined using the formula for single proportion [14]. The prevalence of a cardiovascular risk factor (generalized obesity) among automobile mechanic used in this calculation was 5.8% based on report from a previous study [15]. The confidence interval was taken as 95% and the power of the study was 80%. The minimum sample size for this study was 92 after including 10% attrition. A total of 182 automobile mechanics and 91 age and sex matched controls who were not automobile mechanics were consecutively recruited in the study.

### Study procedure

Questionnaire was administered by the researchers to study participants to obtain socio-demographic information; occupational history such as number of years of practice; use of protective overall gown during work; occupational practices relating to the use of petrochemicals; exposure to petrol and diesel; medical history; and social history such as smoking and alcohol use. Interview was conducted in Yoruba language for the respondents who did not understand English language. All the study participants completed the questionnaires and underwent physical examination.

Height was measured in metres (m) to the nearest 0.1m using a graduated height scale with the participant in erect position without shoes. Weight was measured in kilogram (kg) to the nearest 0.1kg using standard weighing scale with participants wearing light clothing without shoes, cap or headgear. Body mass index (BMI) was calculated as follow;

BMI (kg/m$^2$) = weight (kg)/height$^2$(m$^2$).

The waist circumference was measured using inelastic tape to the nearest 0.1cm, in the horizontal plane, mid-way between the anterior margin of the lowest rib and the iliac crest with the participant standing comfortably and at the end of normal expiration. Blood pressure was measured using Accoson mercury sphygmomanometer. This was taken with participant comfortably seated with arm rested on a table.

Ten ml of fasting blood samples was collected from the study participants for serum lead, cadmium, fasting serum lipid profile, uric acid, total antioxidant capacity (TAC), neutrophil gelatinase-associated lipocalin (NGAL), creatinine and blood glucose. Five ml of spot urine

was collected and analyzed for albumin-creatinine ratio (ACR). ACR was determined using appropriate spectrophotometric methods [16]. Heavy metals levels was determined using atomic absorption spectroscopy.

**Blood sample preparation and digestion for heavy metal analysis.** A modification of the procedure used by Uddin et al [17] was adopted for the digestion of the blood sample. 3 ml of whole blood sample was placed in a test tube, and a 5 ml mixture of nitric-hydrochloric acid ($HNO_3$–HCl) was added to the test tube in a ratio 3:1. The test tube was heated on a hot plate at 100˚C for 3 hours until a clear sample was obtained. After cooling, the sample was then filtered using a 0.45 μm filter paper. The filtrate was made up to 10 mL with deionized water and stored for elemental analysis using an Atomic Absorption Spectrophotometer (Buck 200 model).

TAC was estimated using standard spectrophotometric method based on the ferric reducing antioxidant power (FRAP) assay [18]. 1.5 mL of freshly constituted solutions of 2,4,6-tri-pyridyl-s-triazine and ferric chloride hexahydrate were made to react with 50μL of sample at 37˚C and the result expressed in μmol Trolox equiv/L. Malondialdehyde (MDA) was measured spectrophotometrically based on its reaction with 2-thiobarturic acid (TBA) in acidic pH. This reaction was measured and MDA value calculated [19]. Plasma NGAL was analyzed using Sandwich-ELISA kit produced by the Elabscience US.

**Definition of terms.** Central obesity was defined as waist circumference ≥ 102cm for males [20]. Generalized obesity was defined as BMI ≥30 kg/m$^2$ [21]. Elevated total cholesterol was defined as TC > 5.17mmol/l; low high-density lipoprotein cholesterol (HDL-C) was defined as HDL-C < 1.03mmol/l; elevated low-density lipoprotein cholesterol (LDL-C) was defined as LDL-C >3.40mmol/l; and elevated triglyceride (TG) was defined as TG >1.70mmol/l [22]. Normoalbuminuria was defined as ACR value <30mg/g, microalbuminuria was defined as ACR value between 30-299mg/g and macroalbuminuria was defined as ACR value ≥300mg/g [23]. Reduced glomerular filtration rate (GFR) was defined as a GFR of less than 60mls/mins/1.73m$^2$ [23]. Elevated serum cadmium was taken as cadmium level above the permissible concentration of 0.03–0.12 μg/dl or 0.0003–0.0012 mg/L [24]. Elevated serum lead was taken as lead level greater than the allowed concentration of 0–10 $\mu$g/dL or 0–0.1 mg/L [25]. The atherogenic index of plasma (AIP) was defined as the logarithm of the ratio TG and HDL-C (log TG/HDL-C) [26]. AIP value >0.24 was defined as high cardiovascular risk [26].

**Ethical consideration.** Informed consent was obtained from all participants and the information was treated with utmost confidentiality. Ethical approval with reference number UNIMED/HREC/OndoSt/22/06/21 was obtained from the Human Research and Ethics Committee of the University of Medical Sciences, Ondo State.

**Statistical analysis.** Data obtained were entered and analyzed using the Statistical Package for Social Sciences (SPSS) version 21.0 software 9 (IBM-SPSS, Armonk, NY: IBM Corporation). Descriptive data were presented as tables and categorical variables of the two groups (automobile mechanics and controls) were expressed as proportions and percentages. Normally distributed data were presented as mean and standard deviation while skewed data were presented as median and interquartile range. Association between categorical variables was analyzed using Chi-square. Fisher's exact test was used when the number of counts was less than 5. Correlation was used to determine association between continuous variables. The level of significance for each test was set at $p < 0.05$.

## Results

There were 243 male participants in this study comprising of 162 auto-mechanics and 81 age and sex matched controls. The mean age of the automobile mechanics and the controls were

47.27±9.99 years and 48.94±10.34 years, respectively. One-hundred and thirteen (69.8%) of the automobilemechanics were between the age group of 40–60 years and Christians. Majority (92%) of the automobile mechanics were married. The controls were significantly more educated (p = <0.001) than the automobilemechanics. Ninety-seven (60%) of the automobile mechanics had below 10 years working experience Table 1.

While at work, about 50% of the auto-mechanics regularly wore overall coat, 139 (85.8%) frequently used petrol and diesel to wash their hands and feet and 117 (72.2%) regularly sucked petrol with mouth Fig 1.

The prevalence of elevated TC (32.1% vs 18.5%; p = 0.017), hyperuricemia (20.4% vs 1.2%; p = <0.001), elevated blood glucose (16.0% vs 4.9% p = 0.013); and alcohol use (55.1% vs 30.0%; p = 0.001) were significantly higher in the automobilemechanics compared to the control group. A significantly higher proportion of automobile mechanics had high serum cadmium level compared to the controls (25.3% vs 7.4%; p = <0.001). There was no significant difference in the proportion of automobile mechanics and controls with high serum lead level (82.7% vs 77.8%; p = 0.354) Table 2.

The mean serum TC (4.67±0.90 mmol/l vs 4.41±1.01 mmol/l; p = 0.009); TG (1.37±0.40 mmol/l vs 1.25±0.36 mmol/l; p = 0.021), LDL (3.00±0.79 mmol/l vs 2.74±0.90 mmol/l; p = 0.032), blood glucose (6.30±3.13 mmol/l vs 5.76±0.96 mmol/l; p = 0.027) and uric acid (5.59 ±1.79 mg/l vs 3.81 ±1.59; p = <0.001) were significantly higher in the automobile mechanics. The median values of NGAL in the automobile mechanics and controls were 6.04

**Table 1. Socio-demographic characteristics of study participants.**

| Variables | Automobile Mechanic Group n = 162 | Control Group n = 81 | p-value |
|---|---|---|---|
| | Frequency (%) | Frequency (%) | |
| Age | | | |
| Mean Age | 47.27±9.99 | 48.94±10.34 | |
| <40 years | 27(16.7) | 13(16.0) | 0.992 |
| 40–60 years | 113(69.8) | 57(70.4) | |
| ≥61 years | 22(13.5) | 11(13.6) | |
| Gender | | | |
| Male | 162(100) | 81(100) | |
| Marital Status | | | |
| Single | 10(6.2) | 6(7.4) | 0.915 |
| Married | 149(92.0) | 74(91.4) | |
| Divorced | 3(1.8) | 1(1.2) | |
| Religion | | | |
| Christianity | 113(69.8) | 80(98.6) | <0.001 |
| Islam | 48(29.6) | 1(1.4) | |
| Others | 1(0.6) | 0(0) | |
| Educational Level | | | |
| None | 3(1.9) | 1(1.4) | <0.001 |
| Primary | 55(33.9) | 6(7.1) | |
| Secondary | 92(56.8) | 21(25.7) | |
| Tertiary | 12(7.4) | 53(65.7) | |
| Working Experience | | | |
| <10 years | 97(59.9) | | |
| 10–20 years | 38(23.5) | | |
| >20 years | 27(16.7) | | |

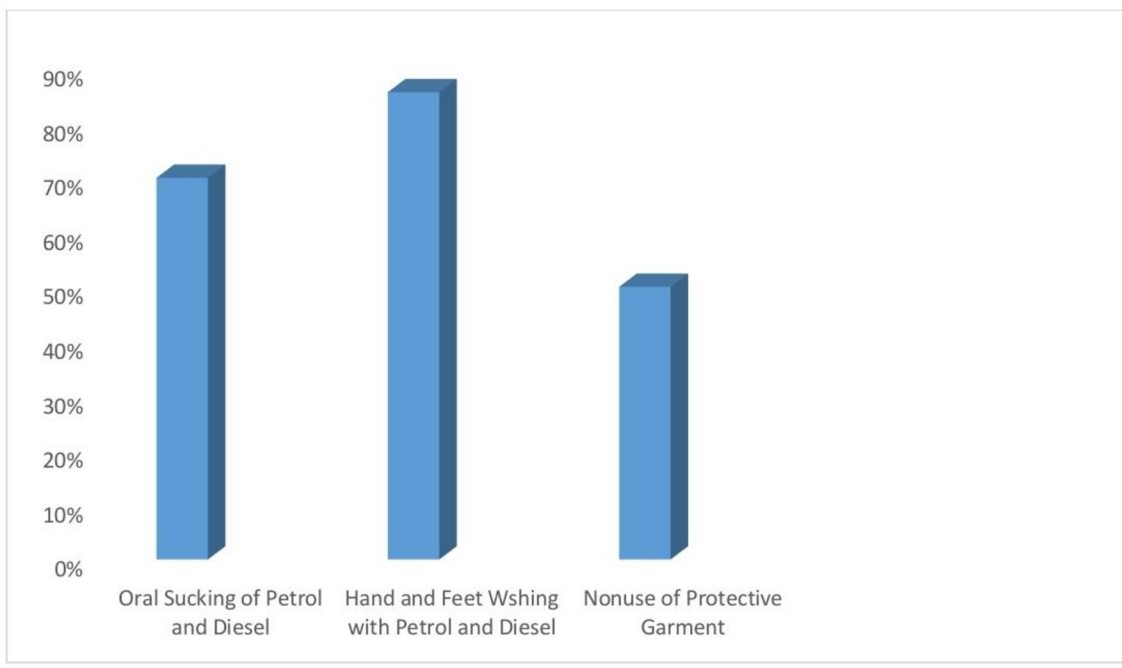

**Fig 1. Harmful occupational practices among automobile mechanics.**

(4.39–6.85) ng/ml and 5.91(4.88–7.24) ng/ml respectively. The median values of TAC in the automobile mechanics and controls were 392.40(228.65–510.55) μmol Trolox equiv/L and 424.50(304.4–592.70) μmol Trolox equiv/L respectively. There was no significant difference in the median values of the TAC and NGAL between the two groups Table 3.

Among the automobile mechanics, there was significant positive correlation between serum cadmium, AIP ($p = 0.024$; $r = 0.382$); and TG ($p = 0.020$; $r = 0.391$). Similarly, there was significant positive correlation between serum lead and NGAL ($p = <0.001$; $r = 0.329$) Table 4.

Among all study participants, there was significant positive correlation between serum cadmium level, AIP ($p = 0.016$; $r = 0.373$) and TG ($p = 0.004$; $r = 0.439$). There was also positive correlation between serum lead and NGAL levels ($p = 0.005$; $r = 0.206$) Table 5.

## Discussion

This study determined some cardiovascular risk factors, kidney function, and their association with heavy metals among automobile mechanics. Automobile mechanics have notable exposure to heavy metals and a higher prevalence of some cardiovascular risk factors, hence they have higher predisposition to cardiovascular disease and kidney disease.

The mean age of the automobile mechanics in this study was 47 years. This is comparable to 44.6 years reported by Saliu et al [27] in a similar study from Nigeria. The mean age of the study participants was lower than 28.6 years reported in a study done in Nepal [28]. All the participants in this study were males which is similar to report of study by Ozomata et al [29]. This is due to the fact that automobile repair work is a male dominant profession in Nigeria.

The proportion of automobile mechanics that regularly used overall protective coat or wear while working was 50%. This is comparable to 49.3%% reported in Nepal [28], but higher than 40% reported by Ozomata et al [29] in a previous Nigerian study and 27% reported among vehicle repair artisans in a study done in Ghana [30]. Majority of the automobile mechanics in this study sucked petrol directly with their mouth and frequently used petrol to wash their

**Table 2. Comparison of cardiovascular risk between the automobile mechanics and controls.**

|  | Auto-mechanic Group n = 162 | Control Group n = 81 | P-value |
|---|---|---|---|
|  | Frequency (%) | Frequency (%) |  |
| Elevated Serum Cadmium | 41(25.3) | 6(7.4) | 0.002 |
| Elevated Serum Lead | 134(82.7) | 63(77.8) | 0.363 |
| High LDL-C | 52(32.1) | 19(23.5) | 0.105 |
| Low HDL-C | 57(35.2) | 31(38.3) | 0.369 |
| High TC | 52(32.1) | 15(18.5) | 0.017 |
| High TG | 30(18.5) | 8(9.9) | 0.093 |
| Hyperuricemia | 33(20.4) | 1(1.2) | <0.001 |
| Elevated Blood Glucose | 26(16.1) | 4(4.9) | 0.013 |
| Hypertension | 57(35.2) | 39(48.2) | 0.070 |
| Obesity/Overweight | 66(40.7) | 33(40.7) | 1.00 |
| Abdominal obesity | 16(9.9) | 8(9.9) | 1.00 |
| Reduce GFR | 43(26.5) | 16(19.8) | 0.157 |
| Albuminuria |  |  |  |
| Normoalbuminuria | 90(55.6) | 45(55.6) |  |
| Microalbuminuria | 68(42.0) | 33(32.7) | 0.870 |
| Macroalbuminiuria | 4(2.5) | 3(3.7) |  |
| Smoking | 33(20.5) | 20(24.7) | 0.234 |
| Alcohol use | 86(53.1) | 21(25.9) | 0.001 |
| Elevated AIP | 21(13.0) | 6(7.40) | 0.279 |

LDL-C (low density lipoprotein-cholesterol); HDL-C (high density lipoprotein-cholesterol); TC (total cholesterol); TG (triglyceride); AIP(atherogenic index of plasma); GFR (glomerular filtration rate)

hands to remove oil residue when repairing vehicles. These unhealthy practices especially sucking of petrol with their mouth exposes majority of them to heavy metals directly by ingestion or indirectly by absorption through the oral mucosa in the course of their work. The findings from this study showed the need for automobile mechanics to be regularly sensitized and health educated on the dangerous effects of exposure to heavy metals as well as the need to

**Table 3. Comparison of mean values of lipid parameters, NGAL, TAC and uric acid between the automobile mechanics and controls.**

| Variable | Automobile mechanic Group n = 162 | Control Group n = 81 | P-value |
|---|---|---|---|
|  | Mean±SD /Median (IQR) | Mean (SD)/Median (IQR) |  |
| TC (mmol/l) | 4.67±0.90 | 4.41±1.01 | 0.009 |
| TG (mmol/l) | 1.37±0.40 | 1.25±0.36 | 0.021 |
| HDL-C (mmol/l) | 1.14±0.24 | 1.09±0.19 | 0.107 |
| LDL-C (mmol/l) | 3.00±0.79 | 2.74±0.90 | 0.032 |
| Blood Glucose (mmol/l) | 6.30±3.13 | 5.76±0.96 | 0.027 |
| NGAL* (ng/ml) | 6.04(4.39–6.85) | 5.91(4.88–7.24) | 0.199 |
| TAC* (μmol Trolox equiv/L) | 392.40(228.65–510.55) | 424.50(304.4–592.70) | 0.555 |
| Uric Acid (mmol/l) | 5.59 ±1.79 | 3.81 ±1.59 | <0.001 |

* Expressed as Median (IQR)

LDL-C (low density lipoprotein-cholesterol); HDL-C (high density lipoprotein-cholesterol); TC (total cholesterol); TG (triglyceride); AIP(atherogenic index of plasma); TAC (total antioxidant capacity); NGAL (neutrophil gelatinase-associated lipocalin)

**Table 4. Correlation between serum heavy metals and some parameters among automobile mechanic.**

| Parameters | | 1 | 2 | 3 | 4 | 5 | 6 | 7 | 8 | 9 | 10 | 11 | 12 |
|---|---|---|---|---|---|---|---|---|---|---|---|---|---|
| Cadmium | 1 | 1.00 | | | | | | | | | | | |
| Lead | 2 | -0.26 | 1.00 | | | | | | | | | | |
| TC | 3 | 0.19 | -0.08 | 1.00 | | | | | | | | | |
| HDL | 4 | -0.14 | 0.07 | .264** | 1.00 | | | | | | | | |
| LDL | 5 | 0.15 | -0.05 | 0.942** | 0.04 | 1.00 | | | | | | | |
| TG | 6 | .391* | -0.15 | .319** | 0.02 | 0.13 | 1.00 | | | | | | |
| AIP | 7 | .382* | -0.13 | 0.09 | -.557** | 0.07 | 0.769** | 1.00 | | | | | |
| NGAL | 8 | -0.30 | .329** | -0.08 | 0.04 | -0.05 | -.173* | -.199* | 1.00 | | | | |
| TAC | 9 | 0.07 | -0.12 | -0.06 | -.290** | -0.01 | 0.10 | .260** | 0.02 | 1.00 | | | |
| Creatinine | 10 | 0.09 | 0.00 | -0.06 | 0.07 | -0.11 | .160* | 0.08 | -0.01 | 0.03 | 1.00 | | |
| GFR | 13 | -0.06 | -0.03 | 0.07 | -0.08 | 0.14 | -.163* | -0.08 | -0.02 | -0.07 | -.906** | 1.00 | |
| Uric Acid | 14 | 0.07 | -0.11 | .330** | 0.10 | .250** | .341** | .236** | -0.07 | -0.08 | 0.07 | -0.01 | 1.00 |

\*. Correlation is significant at the 0.05 level (2-tailed).

\*\*. Correlation is significant at the 0.01 level (2-tailed).

LDL-C (low density lipoprotein-cholesterol); HDL-C (high density lipoprotein-cholesterol); TC (total cholesterol); TG (triglyceride); AIP(atherogenic index of plasma); TAC (total antioxidant capacity); NGAL (neutrophil gelatinase-associated lipocalin), GFR (glomerular filtration rate)

routinely use protective wears and engage in safe and healthy practices when at work, to limit occupational hazards.

There was significantly higher proportion of automobile mechanics with elevated serum levels of cadmium compared to control. This is similar to the findings from previous studies [31–33]. The harmful occupational practices with consequent excessive exposure to heavy metals may be potentially responsible for the elevated levels. Among the automobile mechanics and the controls in this study, no significant difference was found in the proportion of those with elevated serum lead levels. This finding is different from the reports of similar studies that

**Table 5. Correlation between serum heavy metals and some parameters among automobile mechanic and controls.**

| Parameters | | 1 | 2 | 3 | 4 | 5 | 6 | 7 | 8 | 9 | 10 | 11 | 12 |
|---|---|---|---|---|---|---|---|---|---|---|---|---|---|
| Cadmium | 1 | 1.00 | | | | | | | | | | | |
| Lead | 2 | -0.29 | 1.00 | | | | | | | | | | |
| TC | 3 | 0.23 | -0.07 | 1.00 | | | | | | | | | |
| HDL-C | 4 | 0.02 | 0.00 | .302** | 1.00 | | | | | | | | |
| LDL-C | 5 | 0.14 | -0.02 | .949** | 0.10 | 1.00 | | | | | | | |
| TG | 6 | .439** | -.154* | .334** | 0.02 | .183** | 1.00 | | | | | | |
| AIP | 7 | .373* | -0.09 | 0.09 | -.535** | 0.09 | .781** | 1.00 | | | | | |
| NGAL | 8 | -0.28 | .206** | -0.10 | -0.05 | -0.06 | -.252** | -.200** | 1.00 | | | | |
| TAC | 9 | 0.02 | -0.03 | 0.02 | -.228** | 0.05 | .173** | .284** | 0.01 | 1.00 | | | |
| Creatinine | 10 | 0.12 | 0.02 | 0.00 | 0.07 | -0.05 | .171** | 0.10 | -0.05 | 0.11 | 1.00 | | |
| GFR | 11 | -0.10 | -0.03 | -0.02 | -0.10 | 0.03 | -.161* | -0.07 | 0.00 | -.142* | -.905** | 1.00 | |
| Uric Acid | 12 | 0.22 | -0.12 | .310** | 0.12 | .236** | .314** | .195** | -.167** | -0.05 | .158* | -0.10 | 1.00 |

\*. Correlation is significant at the 0.05 level (2-tailed).

\*\*. Correlation is significant at the 0.01 level (2-tail

LDL-C (low density lipoprotein-cholesterol); HDL-C (high density lipoprotein-cholesterol); TC (total cholesterol); TG (triglyceride); AIP(atherogenic index of plasma); TAC (total antioxidant capacity); NGAL (neutrophil gelatinase-associated lipocalin), GFR (glomerular filtration rate)

showed significantly higher serum lead levels in those who were occupationally exposed compared to the control [9, 31, 33]. This finding underscores the need to look beyond occupation as a risk factor for exposure to some heavy metals such as lead. It implies the possible contamination of persons by other potential indirect sources apart from direct contact with petrochemical products. There is need to explore other potential sources of contamination such as soil, water, consumed food items as well as commodities that people are constantly exposed to. This position is strengthened by reports of previous studies that have established higher concentration of heavy metals in soil and water bodies in Nigeria [34–36]. In context, Dix-Cooper et al [37] reported significant association between seafood consumption and the use of herbal remedies with increased serum levels of some heavy metals among Asian women living in Canada. Matouke et al [38] reported high level of heavy metals beyond permissible values in frequently consumed fishes in North Central region of Nigeria. Similarly, Sarkar et al [39] reported higher amount of heavy metal in shrimps from some farms in Bangladesh. Reports from studies by Biose et al [40] and Edogbo et al [35] also showed higher quantities of heavy metals in vegetables, tubers and soil in different parts of Nigeria. The above therefore highlights the need for a holistic approach to ensure the protection of people from both direct and indirect exposure to heavy metals.

Automobile mechanics had significantly higher mean value of total cholesterol and triglyceride compared to the controls in this study. This is similar to the pattern of lipid abnormalities reported in some previous studies among similar study participants [41–43]. There was significant positive association between serum levels of cadmium, triglyceride and AIP in the study participants. This is supported by the finding of the Korean National Environmental Health Survey conducted among 2,519 participants that showed remarkable association between exposure to heavy metals and dyslipidaemia [44]. Similarly, Sharma et al [45] reported significant association between heavy metals and some lipid fractions. Although, studies have reported association between heavy metals and lipid abnormalities, the pathophysiology is not well understood. Overall, the lipid pattern among automobile mechanics suggests that they have higher cardiovascular risk factor compared to the controls. This is similar to findings from previous studies [15, 41–43, 46].

Alcohol consumption was significantly more common among automobile mechanics. About half (53.1%) of them consume alcohol to varying degree. The high prevalence of alcohol use may be due to the fact that artisans such as automobile mechanics habitually consume alcohol based herbal products which are mostly sold or hawked in the their workshops and garages. The frequency of alcohol consumption is similar to 53.4% reported by Akintunde et al. [15]. However, it is higher than 31.2% reported by Ajani et al. [46]. About one-fifth (20.5%) of the automobile mechanics smoked cigarette which is similar to 18.3% reported by Ajani et al, [46] but higher than 10.7% reported by Akintunde et al. [15]. This consumption pattern could predispose them to the adverse cardiovascular and non-cardiovascular consequences of alcohol and tobacco.

The prevalence of hyperuricemia was significantly higher in the automobile mechanics compared to the control. This finding is keeping with report of previous studies conducted within and outside Nigeria [9, 32, 47, 48]. Hyperuricemia is a non-traditional cardiovascular risk factor that is associated with target organ damage, cardiovascular morbidity and mortality [49]. Although the pathophysiology is not fully understood, it has been suggested that uric acid has adverse effect on cardiovascular system by causing endothelial dysfunction, inflammation, increased oxidative stress, insulin resistance, metabolic dysregulation, vasoconstriction and proliferation of vascular smooth muscle [50]. Elevated blood glucose level was also significantly more common in the automobile mechanics. The prevalence rates of these traditional

cardiovascular risk factors ranged between 9.9 and 40.7%. The high prevalence of cardiovascular risk observed in this study is similar to previous reports by Akintunde et al. [15].

A higher proportion of automobile mechanics had low GFR, this was not statistically significant. This finding is similar to report of Oktem et al [10] that showed no significant difference in the levels of serum creatinine and GFR between those who were exposed to heavy metals and those who were unexposed. However, it is different from report by Alasia et al [9] that showed significantly lower GFR and higher serum creatinine in those exposed to heavy metals. NGAL is an early marker of kidney injury and has been found to be highly valuable in both acute kidney injury and chronic kidney disease [51]. There was significant positive association between NGAL, cadmium and lead in this study. These findings are supported by the report of Lentini et al [52] that showed that exposure to heavy metals could cause both glomerular and tubular injury in the kidneys.

The limitation of this study was that causal association between heavy metal, cardiovascular risk factor and kidney function could not be ascertained as this was a cross-sectional study. Secondly, we could not exclude the influence of the use supplements in this study. Thirdly, the findings of this study can also not be generalized due to its relatively small study size. The findings of this study can however, serve as a basis for conducting a large population and longitudinal study to ascertain causality between heavy metal exposure and cardiovascular risk and kidney damage.

In conclusion, this study showed that automobile mechanics have a higher prevalence of elevated total cholesterol, elevated blood glucose, alcohol use and hyperuricaemia and exposure to heavy metals such as cadmium and lead. There was positive association between heavy metals, triglyceride, atherogenic index of plasma and NGAL. Regular health education and sensitization should be conducted to educate artisans on the dangers of exposure to heavy metals together with measures to adopt to significantly limit occupational exposures to heavy metals. Policies and environmental laws that would significantly reduce direct and indirect exposure to heavy metals should be properly implemented in Nigeria. Large scale longitudinal study to determine causality between heavy metals, cardiovascular risk and kidney function should be conducted in Nigeria.

## Supporting information

**S1 Checklist. STROBE statement checklist.**
(DOCX)

## Author Contributions

**Conceptualization:** Oluseyi Ademola Adejumo, Adenike Christianah Enikuomehin, Adeyemi Ogunleye, Walter Bamikole Osungbemiro, Alex Adedotun Adelosoye, Ayodeji Akinwumi Akinbodewa, Olutoyin Morenike Lawal, Kenechukwu Okonkwo.

**Data curation:** Oluseyi Ademola Adejumo, Adenike Christianah Enikuomehin, Adeyemi Ogunleye, Walter Bamikole Osungbemiro, Alex Adedotun Adelosoye, Ayodeji Akinwumi Akinbodewa, Rasheed Olanshile Oloyede.

**Formal analysis:** Oluseyi Ademola Adejumo, Adeyemi Ogunleye, Walter Bamikole Osungbemiro, Ayodeji Akinwumi Akinbodewa, Olutoyin Morenike Lawal, Oladimeji Adedeji Junaid, Kenechukwu Okonkwo, Rasheed Olanshile Oloyede.

**Funding acquisition:** Oluseyi Ademola Adejumo, Kenechukwu Okonkwo.

**Investigation:** Oluseyi Ademola Adejumo, Emmanuel Oladimeji Alli.

**Methodology:** Oluseyi Ademola Adejumo, Adeyemi Ogunleye, Walter Bamikole Osungbemiro, Alex Adedotun Adelosoye, Olutoyin Morenike Lawal, Stanley Chidozie Ngoka, Emmanuel Oladimeji Alli, Rasheed Olanshile Oloyede.

**Supervision:** Oluseyi Ademola Adejumo, Adenike Christianah Enikuomehin, Alex Adedotun Adelosoye, Oladimeji Adedeji Junaid.

**Writing – original draft:** Oluseyi Ademola Adejumo, Stanley Chidozie Ngoka, Oladimeji Adedeji Junaid, Kenechukwu Okonkwo, Emmanuel Oladimeji Alli, Rasheed Olanshile Oloyede.

**Writing – review & editing:** Oluseyi Ademola Adejumo, Alex Adedotun Adelosoye, Stanley Chidozie Ngoka, Oladimeji Adedeji Junaid, Kenechukwu Okonkwo, Rasheed Olanshile Oloyede.

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
