## [Decision Letter · Decision Letter 0]

30 Aug 2023

PONE-D-23-16434Cardiovascular Risk Factors and Renal Function among Auto-mechanic and their association with Serum Heavy Metals: A Cross-sectional StudyPLOS ONE

Dear Dr. Adejumo,

Thank you for submitting your manuscript to PLOS ONE. After careful consideration, we feel that it has merit but does not fully meet PLOS ONE’s publication criteria as it currently stands. Therefore, we invite you to submit a revised version of the manuscript that addresses the points raised during the review process. Please submit your revised manuscript by Oct 14 2023 11:59PM. If you will need more time than this to complete your revisions, please reply to this message or contact the journal office at plosone@plos.org. Please include the following items when submitting your revised manuscript:A rebuttal letter that responds to each point raised by the academic editor and reviewer(s). You should upload this letter as a separate file labeled 'Response to Reviewers'.A marked-up copy of your manuscript that highlights changes made to the original version. You should upload this as a separate file labeled 'Revised Manuscript with Track Changes'.An unmarked version of your revised paper without tracked changes. You should upload this as a separate file labeled 'Manuscript'.If applicable, we recommend that you deposit your laboratory protocols in protocols.io to enhance the reproducibility of your results. Protocols.io assigns your protocol its own identifier (DOI) so that it can be cited independently in the future. For instructions see: https://journals.plos.org/plosone/s/submission-guidelines#loc-laboratory-protocols. Additionally, PLOS ONE offers an option for publishing peer-reviewed Lab Protocol articles, which describe protocols hosted on protocols.io. Read more information on sharing protocols at https://plos.org/protocols?utm_medium=editorial-email&utm_source=authorletters&utm_campaign=protocols.

We look forward to receiving your revised manuscript.

Kind regards,

Olabamiji Abiodun Ajose, MB.BS., M.Sc., MD, FWACP

Academic Editor

PLOS ONE

Journal Requirements:

3. Please amend the manuscript submission data (via Edit Submission) to include author "Adeyemi Ogunleye".

4. Please include a separate caption for your figure in your manuscript.

**Additional Editor Comments:**

You are please requested to attend to all the comments made by the two reviewers.  In addition, you are advised to replace the phrase '"an association between" to "a relationship between" as no statistical test of association was done, but only correlation tests. Also, state whether a particular relationship is positive or negative. 

Reviewers' comments:

Reviewer's Responses to Questions

**Comments to the Author**

1. Is the manuscript technically sound, and do the data support the conclusions?

Reviewer #1: Yes

Reviewer #2: Yes

2. Has the statistical analysis been performed appropriately and rigorously? 

Reviewer #1: Yes

Reviewer #2: Yes

3. Have the authors made all data underlying the findings in their manuscript fully available?

Reviewer #1: Yes

Reviewer #2: Yes

4. Is the manuscript presented in an intelligible fashion and written in standard English?

Reviewer #1: Yes

Reviewer #2: Yes

5. Review Comments to the Author

Reviewer #1: As contained in the the returned manuscript and attached word document. The study is informative and adds to the body of knowledge. It is acceptable after these minor corrections. It was easy to read and it flows. Study population I would like to know how this was achieved i.e. the statement “Any auto-mechanic with known renal, heart, and liver diseases was excluded". [ How was this done? Pre or post sampling?]

How did they exclude the use of supplements like herbal teas that enables detoxifications from being responsible for the results we are seeing.

It would also be interesting to look at the lead and cadmium levels according to years of experience of the auto mechanics, an interesting result may come out here.

Reviewer #2: Dear Editor-in-Chief,

Thank you for the priviledge given to review the manuscript: Cardiovascular Risk Factors and Renal Function among Auto-mechanic and their association with Serum Heavy Metals: A Cross-sectional Study.

Adejumo and colleagues have endeavored to evaluate the possible relationship(s) between serum heavy metals (lead and cadmium) and cardiovascular risk factors, and renal function among auto-mechanic which is a worthwhile adventure.

Below are my observations, comments and suggestions:

Abstract:

Introduction:

The sentence: This study determined some cardiovascular risk factors and renal function among auto-mechanic and their association with heavy metals among automobile mechanics.

It may not be necessary to repeat the word auto-mechanic (i.e automobile: mechanic). Thus, the sentence could be simply put as:

This study determined some cardiovascular risk factors, renal function, and their association with heavy metals among automobile mechanics.

OR:

This study determined some cardiovascular risk factors and renal function among auto-mechanic and their association with heavy metals

Method:

Associations between serum lead, cadmium and some cardiovascular risks were determined.

Results

There were significant positive correlations between serum cadmium level.

Conclusion:

Auto-mechanics have notable exposure to heavy metals and they have a higher prevalence of cardiovascular some risk factors.

Auto-mechanics have notable exposure to heavy metals and they have a higher prevalence of some cardiovascular risk factors.

Introduction:

Line 68: hypertension, obesity, diabetes mellitus, tobacco use, dyslipidaemia, obesity, depression, environmental

The word obesity is listed twice, one should be deleted.

Methodology

Line 126: Heavy metals levels was were determined using atomic spectrometer (atomic absorption spectrophotometry?)

Lines 142 -145:

Definition of terms

142……………………………..Elevated total cholesterol was defined as TC >200 mg/dl; low high-density

143 lipoprotein cholesterol (HDL-C) was defined as HDL-C < 40 mg/dl; elevated low-density lipoprotein

144 cholesterol (LDL-C) was defined as LDL-C >130 mg/dl; and elevated triglyceride (TG) was defined as TG 145 >150 mg/dl.2

The units of measurement here are all conventional units; I suggest that the values be also stated for S. I units (in brackets) for the sake of readers who may be used to mainly S.I. units and not conventional units, except if the journal style allows the use of conventional units only.

Ethical consideration:

State the ethical approval number, if required by the journal style.

Results

Lines 170: Majority (state the percentage of this majority) of the study participants were between the age group of 40-60 years, Christians, and married.

Line 173: While at work, about 50% (state the exact percentage) of the auto-mechanics regularly wore overall coat, 156

You can use statetements like: about half of the auto-mechanics (then state the exact percentage in bracket immediately after)

Table 3:

Serum Blood Glucose (mmol/l) 6.30±3.13 5.76±0.96 0.027

This should be simply stated as blood glucose (if whole blood was used as the sample), or plasma glucose, or serum glucose depending on the specific sample used for glucose assay.

Serum NGAL* (ng/ml) 6.03 (2.47) 5.68 (2.10) 0.048

Serum TAC* (in μmol Trolox equiv/L. 392.40 (221.90) 424.50 (288.3) 0.555

For better understanding, authors should state interquartile range (IQR) as a range (Q1 – Q3), and NOT as a whole figure which is the difference between Q3 and Q1.

Discussion:

Lie s196 – 199:

196 The study found that the auto-mechanics had significantly higher

197 prevalence of some of the cardiovascular risk factors as well as elevated serum levels of cadmium

198 compared to the control group who were not auto-mechanics. In addition, a positive association was

199 found between serum levels of some heavy metals, cardiovascular risk factors and renal function.

The facts in these lines have already been overtly stated in the results section; discussion should therefore entail the explicit implications of the findings, and compare with other reports, stating the similarities and differences with possible reason/s for eac of them.

Line 254: About 21% of them smoked cigarette which is similar to 18.3% reported by Ajani

Percentage is a specific figure: the above can be phrased as: About one-fifth (21%), if 21 is the actual percentage, otherwise authors should state the actual percentage in bracket, which may be slightly less than or greater than 21%.

Conclusion:

Lines 288 and 289: There was an association between serum lead and cadmium

levels and some cardiovascular risk factors as well as markers of kidney damage

Conclusion is the take home message (the final inference) of a study, authors should state specifically which cardiovascular risk factors were found to be associated with serum lead and cadmium levels in the study.

.

6. PLOS authors have the option to publish the peer review history of their article (what does this mean?). If published, this will include your full peer review and any attached files.

Reviewer #1: No

Reviewer #2: No

---

## [Author Response · Author response to Decision Letter 0]

5 Sep 2023

Reviewer 1 Authors’ Response

It may not be necessary to repeat the word auto-mechanic (i.e automobile: mechanic). Thus, the

sentence could be simply put as:

This study determined some cardiovascular risk factors, renal function, and their association with

heavy metals among automobile mechanics.

 Corrected

Method:

Associations between serum lead, cadmium and some cardiovascular risks were determined.

 Corrected

Results

There were significant positive correlations between serum cadmium level.

 Corrected

Conclusion:

Auto-mechanics have notable exposure to heavy metals and they have a higher prevalence of

cardiovascular some risk factors.

Auto-mechanics have notable exposure to heavy metals and they have a higher prevalence of

some cardiovascular risk factors.

 Corrected

Introduction:

Line 68: hypertension, obesity, diabetes mellitus, tobacco use, dyslipidaemia, obesity,. The word obesity is listed twice, one should be deleted.

 Corrected

Methodology

Line 126: Heavy metals levels was were determined using atomic spectrometer (atomic

absorption spectrophotometry?)

 Corrected

Lines 142 -145:

Definition of terms

The units of measurement here are all conventional units; I suggest that the values be also stated

for S. I units (in brackets) for the sake of readers who may be used to mainly S.I. units and not

conventional units, except if the journal style allows the use of conventional units only.

 Variables expressed in S.I unit (mmol/l)

Ethical consideration:

State the ethical approval number, if required by the journal style. Protocol number provided

Lines 170: Majority (state the percentage of this majority) of the study participants were between

the age group of 40-60 years, Christians, and married.

 More details provided

While at work, about 50% (state the exact percentage) of the auto-mechanics regularly

wore overall coat, 156

You can use statements like: about half of the auto-mechanics (then state the exact percentage

in bracket immediately after)

 Revised accordingly

Table 3:

Serum Blood Glucose (mmol/l) 6.30±3.13 5.76±0.96 0.027

This should be simply stated as blood glucose (if whole blood was used as the sample), or

plasma glucose, or serum glucose depending on the specific sample used for glucose assay.

 Serum deleted as advised

Line 173: 

Serum NGAL* (ng/ml) 6.03 (2.47) 5.68 (2.10) 0.048

Serum TAC* (in μmol Trolox equiv/L. 392.40 (221.90) 424.50 (288.3) 0.555

For better understanding, authors should state interquartile range (IQR) as a range (Q1 – Q3),

and NOT as a whole figure which is the difference between Q3 and Q1. IQR provided as Q1-Q3

Line 191-196

The facts in these lines have already been overtly stated in the results section; discussion should

therefore entail the explicit implications of the findings, and compare with other reports, stating

the similarities and differences with possible reason/s for eac of them. Opening paragraph revised. The results part deleted

Line 254: About 21% of them smoked cigarette which is similar to 18.3% reported by Ajani

Percentage is a specific figure: the above can be phrased as: About one-fifth (21%), if 21 is the

actual percentage, otherwise authors should state the actual percentage in bracket, which may be

slightly less than or greater than 21%.

 Corrected as suggested

Conclusion:

Lines 288 and 289: There was an association between serum lead and cadmium

levels and some cardiovascular risk factors as well as markers of kidney damage

Conclusion is the take home message (the final inference) of a study, authors should state

specifically which cardiovascular risk factors were found to be associated with serum lead and

cadmium levels in the study. Corrected as suggested

REVIEWER 2

Tittle: This phrase “A Cross-sectional Study” should be deleted and supplied under material and

methods.. However, it is helpful to have an idea where the study was carried out as part of the title. Corrected as suggested

Lines 22-24. For the short running title, I suggest “Cardiovascular risk factors and renal function in

auto mechanics in association with serum heavy metals.

.

 Corrected as suggested

In Abstract Line 39, There was a repetition in the sentence which should be corrected Corrected as suggested

Introduction. Line 88. The authors did not clearly justify the research gap for this study. This should be

highlighted in the introduction.

 Justification provided

Lines 102 -103. Study population I would like to know how this was achieved i.e. the statement “Any auto-

mechanic with known renal, heart, and liver diseases was excluded". [ How was this done? Pre or post

sampling?] Screening was done by taking history and conducting physical examination pre-sampling

It would also be interesting to look at the lead and cadmium levels according to years of experience of the

auto mechanics, an interesting result may come out here.

 This was not significant. Possibly a large population study follow study will be more revealing

How did we exclude the use of supplements like herbal teas that enables detoxifications from being

responsible for the results we are seeing.

 We did not exclude this. We have included it as part of limitation of the study 

Lines131-132. Please state or reference method used for ACR This has been provided

Results Table 3. The repetition of the word serum should be removed having been initially stated in the

methodology section.

 Done

Discussion. Lines 203-206 This is a rehearse of the results again it is not necessary under discussion.

What is needed here is the implications of the findings. Please comply with this Done

The discussion was lengthy on what was not present in this study among the automobile workers. The oral

sucking of petrol presupposes ingestion from oral mucosa which could lead to elevated cadmium of lead.

This should have been emphasised. The discussion has been revised by deleting the aspect of what is not present among the automobile workers

---

## [Editor Report · Decision Letter 1]

18 Sep 2023

Cardiovascular Risk Factors and Kidney Function among Automobile mechanics and their association with Serum Heavy Metals in Southwest Nigeria

PONE-D-23-16434R1

Dear Dr. Adeyemo

We’re pleased to inform you that your manuscript has been judged scientifically suitable for publication and will be formally accepted for publication once it meets all outstanding technical requirements.

Kind regards,

Olabamiji Abiodun Ajose, MB.BS., M.Sc., MD., FWACP.

Academic Editor

PLOS ONE
---

## [Editor Report · Acceptance letter]

2 Oct 2023

PONE-D-23-16434R1 

Cardiovascular Risk Factors and Kidney Function among Automobile mechanic and their association with Serum Heavy Metals in Southwest Nigeria: a cross-sectional study 

Dear Dr. Adejumo:

I'm pleased to inform you that your manuscript has been deemed suitable for publication in PLOS ONE. Congratulations! Your manuscript is now with our production department. 

Kind regards, 

on behalf of

Professor Olabamiji Abiodun Ajose 

Academic Editor

PLOS ONE